# The Novel Role of *Zfp296* in Mammalian Embryonic Genome Activation as an H3K9me3 Modulator

**DOI:** 10.3390/ijms241411377

**Published:** 2023-07-12

**Authors:** Lu Gao, Zihan Zhang, Xiaoman Zheng, Fan Wang, Yi Deng, Qian Zhang, Guoyan Wang, Yong Zhang, Xu Liu

**Affiliations:** 1College of Veterinary Medicine, Northwest A&F University, Xianyang 712100, China; gaolu1201@163.com (L.G.); zzh1993@nwsuaf.edu.cn (Z.Z.); xiaomanz2017@nwafu.edu.cn (X.Z.); reiserw@nwafu.edu.cn (F.W.); dengyi0246@163.com (Y.D.); zhangyx12212021@163.com (Q.Z.); wangguoyan2022@163.com (G.W.); 2State Key Laboratory for Biology of Livestock, Northwest A&F University, Xianyang 712100, China

**Keywords:** EGA, *Zfp296*, C2H2 domains, H3K9 methylation

## Abstract

The changes in epigenetic modifications during early embryonic development significantly impact mammalian embryonic genome activation (EGA) and are species-conserved to some degree. Here, we reanalyzed the published RNA-Seq of human, mouse, and goat early embryos and found that *Zfp296* (zinc finger protein 296) expression was higher at the EGA stage than at the oocyte stage in all three species (adjusted *p*-value < 0.05 |log2(foldchange)| ≥ 1). Subsequently, we found that *Zfp296* was conserved across human, mouse, goat, sheep, pig, and bovine embryos. In addition, we identified that ZFP296 interacts with the epigenetic regulators KDM5B, SMARCA4, DNMT1, DNMT3B, HP1β, and UHRF1. The Cys_2_-His_2_(C2H2) zinc finger domain TYPE2 TYPE3 domains of ZFP296 co-regulated the modification level of the trimethylation of lysine 9 on the histone H3 protein subunit (H3K9me3). According to ChIP-seq analysis, ZFP296 was also enriched in *Trim28*, *Suv39h1*, *Setdb1*, *Kdm4a*, and *Ehmt2* in the mESC genome. Then, knockdown of the expression of *Zfp296* at the late zygote of the mouse led to the early developmental arrest of the mouse embryos and failure resulting from a decrease in H3K9me3. Together, our results reveal that *Zfp296* is an H3K9me3 modulator which is essential to the embryonic genome activation of mouse embryos.

## 1. Introduction

During early embryogenesis, embryonic genome activation (EGA) is the first critical event in early embryonic development [1,2,3]. After EGA occurs, embryonic development is completely controlled by the zygotic genome, and totipotency is established in the embryos [4]. The starting time of EGA in different species and the changes in biparental transcript and protein levels during the process, as well as the changes in epigenetic modification levels, have received extensive attention [5]. Previous studies have shown that the major EGA in mice occurs at the 2-cell stage and that the minor EGA occurs at the zygote stage [6]; human EGA begins at the one-cell stage and occurs mainly at the 8-cell stage [7], while bovine and goat EGA occurs between the 8- and 16-cell stages [8,9]. Numerous studies have shown that the block of EGA results in early embryonic developmental arrest and the failure of embryonic implantation. Therefore, discovering the roles of key regulators in early embryonic development is fundamental to revealing the system of the embryonic developmental process [10,11]. Although the timing and the activation of embryonic transcription vary across species, they are still conserved and specific to a certain degree in some of the changes in epigenetic events [12,13].

During EGA, different types of histone post-translational modifications play different important roles. During the process of EGA, embryos display changes in a series of epigenetic events, including gene-wide DNA methylation, post-translational modifications of histones, and chromatin remodeling [14,15,16]. H3K9me3, as one of the markers of heterochromatin, is essential for mammalian embryonic development and cell fate determination; a high level of H3K9me3 will arrest early embryonic development. In mouse embryos, the distribution of H3K9me3 in the promoter region is different from that of H3K4me3, showing a negative correlation and a positive correlation with traditional repressive modifications. Promoter regions with high levels of H3K9me3 modification in MII cells rapidly decrease H3K9me3 modification levels in these regions after fertilization, and it turns out that the majority of the genes in these promoter regions are developmental genes that are activated upon zygotic genome activation [17,18,19]. Human embryos have a striking divergence from mouse embryos and distinct species-specific features exist: 8-cell-specific H3K9me3 is temporarily established in enhancer-like regions, and DUX and multiple Krüppel-associated box domain zinc finger proteins (KRAB-ZNFs) have been identified as potential factors for establishing 8C-specific H3K9me3. In addition, at the 8-cell stage in humans, SVA and other hominin-specific reverse transcription transposons begin to undergo massive H3K9me3 demethylation and become activated enhancers, thus promoting major ZGA gene expression [20,21]. In somatic cell nuclear transfer (SCNT) in bovine embryos and goat-cloned embryos, the abnormally higher level of H3K9me3 might serve as an epigenetic barrier to reprogramming [22,23].

A portion of C2H2 proteins is associated with H3K9me3 [24]. For example, ZNF644 and ZNF803/WIZ physically bind the G9a/GLP and together regulate H3K9 methylation [25,26,27]. *Zfp296* is a member of the zinc finger protein family, which encodes a conserved mammalian factor belonging to the Cys_2_/His_2_-type zinc finger (C2H2-ZF) family, and it was identified in leukemia mouse models, ESC, and sperm [28,29,30]. *Zfp296* was shown to be part of a pluripotent-specific transcriptional network as well as an ESC-specific effector that could interact with the NuRD [31,32,33]. Here, we aimed to find a gene that was functionally conserved in the early embryonic development of different mammals, and we hypothesized that the gene was associated with the changes in epigenetic modifications. Finally, we conducted bioinformatic analyses, cell and embryo culture, and molecular biology techniques and showed that *Zfp296* could affect the level of H3K9me3 during the process of EGA. 

## 2. Results

### 2.1. Zfp296 Is Significantly Expressed in the EGA Stage of Human, Mouse, and Goat, and Has High Species Conservation

In order to find a gene that is essential for mammalian embryonic genome activation, we analyzed the wild-type part of RNA-Seq from previous studies [19,34,35] of human, mouse, and goat early embryos to identify differential genes between the oocyte stage and the EGA stage. Comparing the oocyte stage to the EGA stage, humans have 2550 upregulated genes and 3561 downregulated genes, mice have 2520 upregulated genes and 3228 downregulated genes, and goats have 2138 upregulated genes and 2595 downregulated genes (Appendix A). Moreover, we found that humans, mice, and goats shared 29 significantly upregulated genes from the oocyte to the EGA stage. Then, we found that *Zfp296* was significantly upregulated at the EGA stage in all three species when compared to these genes at the oocyte stage (Figure 1A). *Zfp296* was more significantly upregulated in the mid-2-cells, especially in mouse embryos. 

Then, we aligned the DNA sequence and protein sequences of ZFP296 from six species (human, mouse, goat, sheep, bovine, and pig). The results showed a high degree of homology and similarity in *Zfp296*. Furthermore, by Batch-CD-search of NCBI (Figure 1B–D), we found that all six of these species had C2H2 domains. This suggests that the function of ZFP296 may be conserved in the different mammalian EGAs.

Overall, these results demonstrate that *Zfp296* is more likely to have a critical role at the EGA stage of humans, mice, and goats and that it is highly conserved in mammals.

### 2.2. ZFP296 Localizes to the Nucleus and Interacts with Multiple Proteins Which Were Key to Methylation

To further understand the *Zfp296*, we first showed that ZFP296 is localized in the nucleus of F9 cells (Figure 2A). Next, we analyzed the published RNA-Seq of *Zfp296* knockout in mESCs to compare it with the differential genes of EGA in mice [32]; we found that the expression of a subset of genes was altered both during the EGA period and in the *Zfp296* knockout (KO) mESCs (Appendix A); in addition, some of them, such as *Ncoa6*, *Zp3*, *Dazl*, *Eef2k*, *Igf2bp2* (*Imp2*), etc., were found to play an important role in oocyte or early embryonic development [36,37,38,39] (Figure 2B). A further gene ontology (GO) analysis of these differentially expressed genes revealed that a large part of the differential genes after knocking out *Zfp296* were related to methylation and methyltransferase activation (Figure 2C).

To validate the performed analysis, we proved the protein interaction between ZFP296 and KDM5B(lysine demethylase 5B); SMARCA4(SWI/SNF-related, matrix-associated, actin-dependent regulator of chromatin, subfamily a, member 4); DNMT1(DNA methyltransferase 1); DNMT3B(DNA methyltransferase 3 beta); HP1β(chromobox 1); and UHRF1(ubiquitin-like with PHD and ring finger domains 1), which are associated with histone methylation [40,41,42,43,44], (Figure 2D–I) in HEK-293T cells by co-immunoprecipitation. Meanwhile, in early embryogenesis, *Kdm5b* is dramatically increased before the EGA stage; by comparison, *Dnmt1*, *Dnmt3b*, *Hp1β*, and *Uhrf1* have different degrees of downward reduction; *Uhrf1* is the most significantly downregulated before EGA (Appendix A, Appendix A).

In summary, some of the genes regulated by *Zfp296* in embryonic stem cells are related to early embryonic development and methylation, and ZFP296 interacts with KDM5B, SMARCA4, DNMT1, DNMT3B, HP1β, and UHRF1.

### 2.3. C2H2-TYPE2 and C2H2-TYPE3 of ZFP296 Can Co-Regulate H3K9me3

ZFP296 has six ZF domains (Figure 3A), and its C2H2-TYPE2 and C2H2-TYPE3 domains determine the localization of ZFP296 in heterochromatin [45]. As H3K9 methylation is involved in heterochromatin formation, the silencing of heterochromatin genes, and the process of embryonic genome activation [18,46], we downloaded and analyzed the published ChIP-seq of ZFP296, H3K9me3, and H3K4me3 to investigate the link between *Zfp296* and H3K9me3 [32,47].

The ChIP-Seq revealed that ZFP296 was enriched in promoters and was more similar to H3K4me3 (Figure 3B); however, H3K9me3 was enriched in distal intergenic in the mESC genome. Moreover, *Zfp296* was also enriched in *Trim28*, *Suv39h1*, *Setdb1*, *Kdm4a*, and *Ehmt2*, and the mRNA level of these genes also showed significant changes during EGA in humans, mice, and goats (Appendix A). Interestingly, however, their expression did not change in Zfp296 KO mESCs (Appendix A). Next, we transfected four different vectors, which were inserted with full-length *Zfp296* and its mutants (*Zfp296*-Δ2, *Zfp296*-Δ3, and *Zfp296*-Δ2&Δ3; Δ means that the C2H2 domain was deleted.) into F9 cells to explore the effects of C2H2-type2 and C2H2-type3 on H3K9me3 (Figure 3D); an empty vector (e.v.) served as a negative control. The results showed that H3K9me3 was not significantly reduced in comparison to the control group when only C2H2-type2 or C2H2-type3 or both was removed. In contrast, H3K9me3 in the Zfp296 full-length group was more significantly reduced than it was in the other groups (Figure 3E). Therefore, C2H2-type2 and C2H2-type3 co-regulate the modification level of H3K9me3.

To summarize, the C2H2-type2 and C2H2-type3 domains of ZFP296 can co-reduce the modification level of H3K9me3.

### 2.4. Interfering with Zfp296 Expression Arrests Early Embryonic Development of Mice

To explore the role of *Zfp296* in early embryonic development, we collected the oocytes to the 8-cell embryos and observed that *Zfp296* was mainly localized in the nucleus (Figure 4A and Appendix A). Subsequently, we interfered with the expression of *Zfp296* with siRNA at the zygote stage and found that after the knockdown of *Zfp296* (Figure 4B) >60% of the embryos were arrested at the 2-cell stage (Figure 4C,D). To verify whether ZFP296 affects embryonic genome activation by regulating H3K9me3 modification levels, we collected late-2-cell embryos from the treatment and negative control groups. The results of immunofluorescence staining showed that the modification level of H3K9me3 in the treatment group with the knockdown of *Zfp296* was significantly higher than that in the control group (Figure 4E).

To further explore the role of *Zfp296* in early embryonic development, we collected the mid-2-cell embryos and observed that *Zfp296* was mainly localized in the nucleus (Figure 4A). Subsequently, we knocked down its expression by microinjecting siRNA at the zygote stage, and Western blot analysis showed that the levels of ZFP296 protein were dramatically decreased in the siRNA-injected embryos (Figure 4B). In the *Zfp296* siRNA-treated group, >60% of the embryos were arrested at the 2-cell stage compared with the negative control (NC) group (Figure 4C,D). Furthermore, the results of immunofluorescence staining showed that the modification level of H3K9me3 in the *Zfp296* siRNA-treated group was significantly higher than that in the negative control group (Figure 4E).

Overall, these results verified that ZFP296 can affect embryonic genome activation by regulating the modification levels of H3K9me3.

## 3. Discussion

### 3.1. Zfp296 May Play a Similar and Important Role in Early Mammalian Embryonic Development

During EGA, a large number of zygotic genes are transcriptionally activated [48], and the discovery of genes that play crucial regulatory roles in this process can deepen the understanding of the mechanism of EGA. Compared with the oocyte stage, *Zfp296* increased significantly during and after the occurrence of EGA in humans, mice, and goats; the increase in the expression level at the mid-2-cell stage was much higher than at the zygotic stage, especially in mice (Figure 1A). In addition, the comparison of the *Zfp296* DNA sequences and the protein sequences of *Zfp296* in different species revealed that *Zfp296* is conserved across species. The mechanisms of EGA in different species usually have a degree of conservation, including the genes that are important for this process such as those of the Dux family [12,49,50]. So, *Zfp296* may be a key for EGA in different species.

The analysis of the *Zfp296* knockout mESC RNA-seq revealed that some of the differential genes, such as *Zp3*, *Dazl*, and *Igf2bp2*, have important effects on germ cells and early embryonic development. *Zp3* is not only a major component of the zona pellucida (ZP) in mammalian oocytes, it also promotes fertilization by recognizing sperm binding and activating the acrosome reaction, which is very important for normal fertility [51,52,53]. *Dazl* is essential for oocyte maturation and early embryonic development; it controls maternal mRNA translation until the zygote genome becomes activated [38]. *Eef2k* is an evolutionarily conserved regulator of protein synthesis that maintains germ cell quality and eliminates defective oocytes, and the reduction in *Eef2k* in mice causes ovarian cell apoptosis and leads to abnormal follicles and the accumulation of defective oocytes at a high reproductive age [39]; the maternal deletion of *Igf2bp2* (*Imp2*) causes early embryonic developmental arrest in vitro at the 2-cell stage [40]. The deletion of *Igf2bp2* leads to the downregulation of *Ccar1* and *Rps14*, both of which are required for early embryonic developmental competence. *Zfp296* regulates the expression of these genes in mESCs (Figure 2B). On the other hand, *Zfp296* was significantly upregulated in human, mouse, and goat early embryos at the onset of EGA. ZNF proteins show widespread binding to the regulatory regions that target a diverse range of genes and pathways [54]. Thus, *Zfp296* may affect early embryonic development in these species.

### 3.2. Zfp296 Can Affect EGA by Negatively Regulating H3K9me3

Some of the differential gene functions regulated by *Zfp296* were enriched in methylation and methylase activation (Figure 2C). Additionally, ZFP296 has six C2H2-ZF domains, such as *Zfp57*, which were shown to help different DNA binding proteins to recognize DNA methylation [55]; they can also recruit histone methylation-related enzymes to make a difference. At the cellular level, we also revealed that ZFP296 has interactions with KDM5B, SMARCA4, DNMT1, DNMT3B, HP1β, and UHRF1. A series of epigenetic events, including genome-wide DNA demethylation, modifications of histones, and chromatin remodeling, occur in embryos during EGA. In this process, KDM5B actively removes the broad H3K4me3 domains; this results in a shift of H3K4me3 from widespread distribution in the oocyte genome to being strongly maintained at the TSSs in the 2-cell embryo, which is critical for EGA [17]. KDM5B knockdown in pigs by Cas13 has also shown that the abundance of H3K27me3 and H3K9me3 is significantly reduced [56]; SMARCA4 (BRG1) is a catalytic subunit of SWI/SNF-related complexes; the SMARCA4-depleted embryo resulted in 2-cell arrest and reduced transcription for ~30% of the expressed zygotic genes [4,57,58]. In previous report, UHRF1 was targeted for DNA methylation maintenance with DNMT1 by binding H3K9me2/3 or by hemimethylation, and the presence of both binding activities ensured high-fidelity DNA maintenance methylation [59]. During embryogenesis, the depletion of maternal UHRF1 leads to global DNA methylation levels in GV oocytes and zygote decrease and early embryonic developmental arrest at the 2-cell stage [45]. The nuclear localization of DNMT1 in oocytes is also dependent on UHRF1; DNMT1 and UHRF1 both mediate de novo methylation in oocytes [45,60]. In early embryos, Dnmt1 maintains DNA methylation during development. In contrast, *Dnmt3a/b* has no significant role in the maintenance of embryonic methylation [61]. *Hp1β* is enriched in maternal central heterochromatin in an H3K9me3-related *Suv39h2*-dependent manner in mouse zygotes, preventing PRC1 from targeting maternal central heterochromatin in order to regulate the mRNA abundance of genes involved in development and proliferation [62,63]. By ChIP-Seq analysis, we also found that ZFP296 was enriched in *Trim28*, *Suv39h1*, *Setdb1*, *Kdm4a*, and *Ehmt2* (Figure 3C). HP1, G9a, SETDB1, and TRIM28, as H3K9me3 chromatin readers and related enzymes, have also established a role for chromatin modifications in the development of designated 2CLCs [64,65]. The depletion of *Ehmt2* and *Trim28* leads to inappropriate gene expression and embryonic death [66]. In oocytes, *Kdm4a* erases H3K9me3 at a broad level. H3K4me3 is crucial for proper ZGA [19]. The progressive expression and regulatory activity of Suv39h enzymes ensure low levels of H3K9me3 after fertilization, and *Suv39h1*-mediated H3K9me3 prevents epigenetic reprogramming in early embryonic development, leading to strong developmental arrest [67,68]. Interestingly, the knockdown of *Zfp296* had no apparent effect on the expression level of these genes in the mESCs (Appendix A, Appendix A). Meanwhile, the distribution level of ZFP296 in the promoter region of the genome was significantly higher than that of H3K9me3 in the mESCs; our results also revealed that the second and third C2H2-ZF domains of ZFP296 can co-regulate H3K9me3 directly at the cellular level, which indicates that ZFP296 does not affect H3K9me3 levels by regulating the expression of these genes; they may co-regulate the level of H3K9me3. The binding sites of the C2H2-ZF proteins are usually enriched with chromatin marks which are specific to promoters, enhancers, or repressor regions, and they can also mediate some interactions. Most C2H2 proteins have the Krüppel-associated box domain (KRAB), which is associated with H3K9me3; some of them, such as ZNF317, ZFP28, and ZNF273, act as activators [24,69,70]. Previous studies have also demonstrated that ZFP296 can bind to NuRD and affect the binding site of NuDR on the genome in mESCs [32]. Therefore, we suggest that ZFP296 binds to proteins such as KDM5B, etc., and performs a similar function to influence the level of H3K9me3 modifications in the promoter region. On the other hand, the mRNA abundances of *Zfp296*, *Kdm5b*, *Dnmt1*, *Dnmt3b*, *Hp1β*, *Uhrf1*, *G9a*, *Setdb1*, *Trim28*, *Kdm4a*, and *Suv39h1* are all fluctuated before or during EGA in early embryos, with the exception of *Smarca4* (Appendix A, Appendix A), implying that ZFP296 and these genes also have complex roles related to the dramatic changes in H3K9me3 in EGA.

Mammalian embryo genome activation starts with the rapid establishment of H3K9me3 in LTRs and the deletion of H3K9me3 marks in the promoter regions, a state that is preserved in pre-implantation embryos [18]. During the pre-implantation embryogenesis, CpG-rich genomic loci of DNA methylation maintenance usually have high H3K9me3 signaling and DNA methylation levels (CHM) [71]. Abnormal levels of H3K9me3 modification cause early embryos to fail to develop normally. We found that knocking down the expression of *Zfp296* at the late one-cell stage would cause a considerable degree of embryonic developmental arrest at the 2-cell stage, and the level of H3K9me3 would be increased. Furthermore, we showed that *Zfp296* interacted with many H3K9-related enzymes in the cells. Therefore, *Zfp296* is likely to be involved in erasing the H3K9me3 mark in the promoter region of the zygotic genome during embryonic genome activation in order to ensure the normal occurrence of EGA. However, the relationship between *Zfp296* and *Kdm5b*, *Dnmt1*, *Dnmt3b*, *Hp1β*, *Uhrf1*, *G9a*, *Setdb1*, *Trim28*, *Kdm4a*, and *Suv39h1* and mechanisms of *Zfp296* affects the level of H3K9me3 modification in the early mammalian embryo and needs further exploration.

## 4. Materials and Methods

### 4.1. Data Analysis

#### 4.1.1. Data Access, Analysis, and Visualization of RNA-Seq and ChIP-Seq from Previous Studies

To find a gene that performed similar functions during early embryonic development in different mammals and to obtain more information about this gene. We downloaded and re-analyzed the wild-type parts of the RNA-seq from mouse (GSE129731) [19], human (GSE36552) [35], and goat early embryos (GSE129742) [34] and mapped them with mm10, GRCh38, and ARS1. The RNA-seq data of *Zfp296*-KO in the mESCs were downloaded with the GEO accession number GSE117288 [32]. The ChIP-Seq dates of ZFP296, H3K4me3, and H3K9me3 in the mESCs were downloaded with the GEO accession numbers GSE117287 (both ZFP296 and H3K4me3) [32] and GSE180006 [47].

The expression calculation and the differential gene expression analysis of the results were analyzed by the Ballgown R package [72]. The *p*-values (two-tailed) were adjusted using the Benjamini and Hochberg false discovery rate. The differential genes were defined as those with at least |log2 (fold changes) ≥ 1.5| between a pair of samples at padj < 0.05 by R package limma and DESeq2. 

The ChIP-seq of ZFP296, H3K4me3, and H3K9me3 in the mESCs reads to the mouse reference genome (mm10) was mapped by Bowtie2. Peak calling was performed with the MACS 2.0 tool. The peak in the region of the chromosomes was visualized by the annotatePeak function of the R package ChIPseeker (TSS = −2000, 2000) and IGV.

#### 4.1.2. GO Enrichment Analysis

R package AnnotationHub was used for GO enrichment analysis. We chose *p* < 0.05 to indicate significance.

#### 4.1.3. DNA Sequence and Protein Sequence Analysis of *Zfp296*

For cross-species conservation investigation, DNA sequences of *Zfp296* were downloaded from the NCBI (human F45076478-F45071500, mouse F19311212-F19314581, goat, F53813073-F53807048, sheep F52542448-F52538527, bovine F52753570-F52749714, pig F51508284-F51504384). These sequences were analyzed by Muscle and visualized by MEGA. The protein sequences of ZFP296 were downloaded from UniProt (https://www.uniprot.org/, accessed on 20 January 2022) (human, UniProtKB Q8WUU4; mouse, UniProtKB Q8WUU4; goat, UniProtKB A0A452DNB1; sheep UniProtKB A0A6P7ES84; bovine, UniProtKB E1BA69; pig, UniProtKB A0A8D1B633). The protein sequences aligned by the multiple alignment of NCBI (https://www.ncbi.nlm.nih.gov/tools/cobalt/cobalt.cgi?LINK_LOC=BlastHomeLink, accessed on 20 March 2023) and the conserved domains of ZFP296 were searched by Batch-CD-search of NCBI (https://www.ncbi.nlm.nih.gov/Structure/bwrpsb/bwrpsb.cgi, accessed on 24 August 2022).

### 4.2. Cell Culture and Transfection

At 37 °C in a humidified 5% CO_2_/95% air incubator, the F9 cells (ATCC, Manassas, VA, USA) were cultured with DMEM (Gibco, #12800-082, Grand Island, NY, USA) supplemented with 10% fetal bovine serum (Biological Industries, #04-001AUS-1A, Kibbutz, Israel), and the HEK-293T cells (ATCC, VA, USA) were cultured with DMEM (Gibco, #12800-082) supplemented with 10% FBS (Gibco, #10100-147) on different 60 mm Petri dishes (Corning Costar, #430166, Corning, NY, USA). The F9 cells and HEK-293T cells with a confluence of 70% to 90% were used for transfection according to the protocol of the Lipofectamine 2000 reagent (Invitrogen, #11668019, Carlsbad, CA, USA). The F9 cells were used to show the cellular localization of ZFP296 and the function of C2H2 domains as their characteristics are close to those of mESCs; the HEK-293T cells were used to show the proteins that interacted with ZFP296.

### 4.3. Construction of Vectors

We constructed 4 different pCMV-myc-*Zfp296* vectors to show the relationship between the C2H2 domains and H3K9me3: (1) a vector with only the C2H2-type2 domain removed; (2) a vector with only the C2H2-type3 domain removed; (3) a vector with both the C2H2-type2 and the C2H2-type3 domains removed; and (4) a *Zfp296* full-length vector. In addition, one pEGFP-C1-Zfp296 vector was constructed to show the cellular localization of ZFP296. All the sequences were amplified from mouse lung cDNA. The primer sequences used are shown in Table 1.

To find out whether ZFP296 could interact with the methylation-related enzyme, the full-length coding sequences of *Kdm5b*, *Smarca4*, *Dnmt1*, *Dnmt3b*, *Hp1β*, and *Uhrf1* were also amplified from mouse lung. The primer sequences of the vectors used for co-immunoprecipitation are shown in Table 2.

### 4.4. Western Blot

After 36 h of cell transfection, RIPA buffer (Beyotime, #P0013B, Shanghai, China) supplemented with a protease inhibitor (Thermo, #87786, Newington, NH, USA) was used to lyse the cells. Five microliters of the samples were removed for protein content determination (Transgen, #DQ111-01, Beijing, China), and 20 μg of proteins was electrophoresed on sodium dodecyl sulphate (SDS)-polyacrylamide gels and transferred to polyvinylidene difluoride membranes (Millipore, #ISEQ00010, Burlington, MA, USA) by semidry transfer for 2 h at 250 mA. After Tris-buffered saline with 0.1% Tween-20 (TBST) blocking for 4 h, the membranes were incubated with a primary antibody overnight at 4 °C. After washing three times with TBST, the membranes were incubated with a horseradish peroxidase (HRP)-conjugated secondary antibody (Beyotime, #A0208, Rabbit and #A0216, Mouse, 1:2000 dilution) for 2 h at room temperature. Images were obtained from the ChemiDoc MP Imaging System (Bio-Rad, Hercules, CA, USA) and analyzed by ImageJ. 

The anti-histone H3 served as a loading control, and the primary antibodies used were anti-Zfp296 (SantaCruz, #sc-514868, Santa Cruz, CA, USA, mouse monoclonal 1:500 dilution); anti-histoneH3 (Proteintech, #17168-1-AP, Wuhan, China, Rabbit 1:500 dilution); and anti-H3K9me3 (ACTIVE&MOTIF, #61014, Carlsbad, CA, USA, mouse monoclonal 1:500 dilution).

### 4.5. Co-Immunoprecipitation

To confirm whether ZFP296 interacts with proteins such as KDM5B, we transfected the *Zfp296*-myc vector or *Zfp296*-Bio vector with other tagged-protein vectors into HEK-293T cells. After 36 h, the lysis of 1 × 10^7^ F9 cells in 500 μL of 1× IP Lysis Buffer (Thermo, #87787) was used to collect the whole-cell extracts for 30 min on ice. After centrifugation, the protein extracts were incubated at 4 °C overnight with Myc mAb (Beyotime, #AF0033, mouse monoclonal 1:500 dilution) or control immunoglobulin G (Beyotime, #P2265), and Pierce Protein A/G Agarose (Thermo, #26162) was used to capture the precipitated proteins for 6 h at 4 °C. Subsequently, the beads were washed three times with IP Lysis Buffer, and the bound proteins were eluted in 5 × SDS loading buffer and analyzed by Western blot.

After clarifying the whole-cell extracts, the experiment was carried out according to Dynabeads MyOne™ Streptavidin C1 (Thermo, #65002). The samples were used for Western blot analysis.

Myc-tagged ZFP296 or Avidin-tagged ZFP296 was immunoprecipitated, and the precipitates were blotted for Myc-tagged ZFP296 or Avidin-tagged ZFP296. The total cell lysate (input) was set as an internal reference for the co-immunoprecipitation assay, and normal mouse IgG served as a negative control.

### 4.6. Immunofluorescence Staining of Cells

To determine the cellular localization of *Zfp296*, we transfected the pEGFP-ZFP296-C1 vector into the F9 cells. After 36 h, we discarded the culture medium and washed the cells three times with PBS. After fixing in immunofluorescent staining fix solution at 4 °C overnight and blocking in immunofluorescence staining blocking buffer at room temperature for 1 h, the samples were incubated with anti-GFP (Beyotime, #AG281 mouse monoclonal, 1:1000 dilution) for 2 h at room temperature, and the nuclei were stained with DAPI (Beyotime) for 5 min. The experiment was carried out according to the immunofluorescence staining kit (Beyotime, #FD008,). The samples were photographed under an inverted fluorescence microscope (Nikon, Tokyo, Japan). 

### 4.7. Collection of Mouse Zygotes

Adult male and female C57BL/6J mice were purchased from the Experimental Animal Center of the Fourth Military Medical University (Xi’an, China). They were maintained on a 12/12 h light/dark cycle and 50–70% humidity with free access to food and water at the Laboratory Animal Facility of the College of Veterinary Medicine, Northwest A&F University.

One hundred and fifty 6–8-week-old female mice were intraperitoneally injected with 10 IU of the pregnant mare serum gonadotropin (Ningbo, China) and 10 IU of human chorionic gonadotropin (Ningbo, China). Adult male mice were placed with the hCG-injected females overnight, and after 16 h of hCG injection, cervical dislocation and execution of the successfully mated female mice was conducted. The oviducts were washed with M2 medium (Sigma, #6A2917, Darmstadt, Germany), and then, the dilated part of the oviducts was punctured to release the zygotes. The cumulus cells around the zygotes were then moved by hyaluronidase (Sigma, #37326-33-3 USA) at 37 °C for 2–3 min. The embryos were cultured at 37 °C with 5% CO_2_ in fresh KSOM medium (Caisson Laboratories, IVL04, Smithfield, VA, USA). Finally, 1-, 2-, 4-, and 8-cell embryos and blastocyst-stage embryos were collected after 2 to 4, 22 to 26, 48 to 50, 60 to 70, and 96 to 100 h of culture, respectively.

### 4.8. Immunofluorescence Staining of Embryos

Oocytes or embryos were fixed in 4% paraformaldehyde in PBS for 30 min at room temperature. After being permeabilized in PBS containing 0.2% Triton X-100 for 15 min and blocked in 5% bovine serum albumin in PBS for 1 h at room temperature, the samples were incubated overnight with mouse monoclonal anti-Zfp296 (SantaCruz, #sc-514868, 1:1000 dilution) and washed in PBS containing 0.2% polyvinylpyrrolidone (PVP). Next, the secondary antibodies (Beyotime, #A0413, Mouse, 1:200 dilution) were incubated with the samples for 1 h at room temperature and washed in PBS/0.2% PVP. DAPI (Beyotime) was stained with the nuclei (Beyotime) for 5 min. Finally, the samples were observed by a Zeiss Axio Observer D1 microscope (Carl Zeiss, Carl Zeiss, Thornwood, NY, USA). Staining intensities were analyzed by ImageJ. The data were normalized with respect to the background levels, and DAPI was normalized to quantify the fluorescence intensity.

### 4.9. Microinjection

The equivalent of 2–5pl of siRNA or the same concentration of negative control interference at a concentration of 20 µM was injected into the zygotes using an Eppendorf (Hamburg, Germany) micro-manipulator under a Leica inverted microscope. After injection, the zygotes were washed and cultured in fresh KSOM medium at 37 °C with 5% CO_2_. The siRNA used to silence *Zfp296* was synthesized by the Shanghai GenePharma Co., Ltd. (Shanghai, China), and the sequence was 5′-GAAGCGUCAACUCCAAACUTT-3′.

### 4.10. Statistical Analysis

All the data are given as means ± SEM, and each experiment was repeated at least three times. The statistical tests included an un-paired one-tailed or two-tailed Student’s *t*-test and a one-way analysis of variance. A *p*-value > 0.05 was considered as not significant (ns); 0.01 < *p* < 0.05 was significant and indicated with one asterisks *; 0.001 < *p* < 0.01 was very significant and indicated with two asterisks **; and 0.0001 < *p* < 0.001 was extremely significant and indicated with three asterisks ***.

## 5. Conclusions

In conclusion, we not only showed that the expression of *Zfp296* was significantly upregulated at the EGA stage of human, mouse, and goat embryos, but also proved that ZFP296 interacts with multiple proteins associated with histone and DNA methylation at the cellular level. Additionally, C2H2-TYPE2 and C2H2-TYPE3 of ZFP296 can decrease the level of H3K9me3. Moreover, *Zfp296* can also regulate the level of H3K9me3 and affect the development of early embryos.

Collectively, identifying and exploring the molecular and epigenetic factors determining inter-transcriptomic and inter-proteomic communication during EGA processes in human, murine, and caprine embryos might contribute to an improvement in the production of high-quality embryos with the use of a wide spectrum of assisted reproductive technologies (ARTs) and experimental embryology techniques (EETs), not only in the above-indicated three species, but also in other mammalian species. The aforementioned ARTs and EETs include such modern methods as in vitro fertilization by either gamete co-incubation or intracytoplasmic sperm injection [73,74,75,76] and somatic cell cloning [77,78,79].

## Figures and Tables

**Figure 1 ijms-24-11377-f001:**
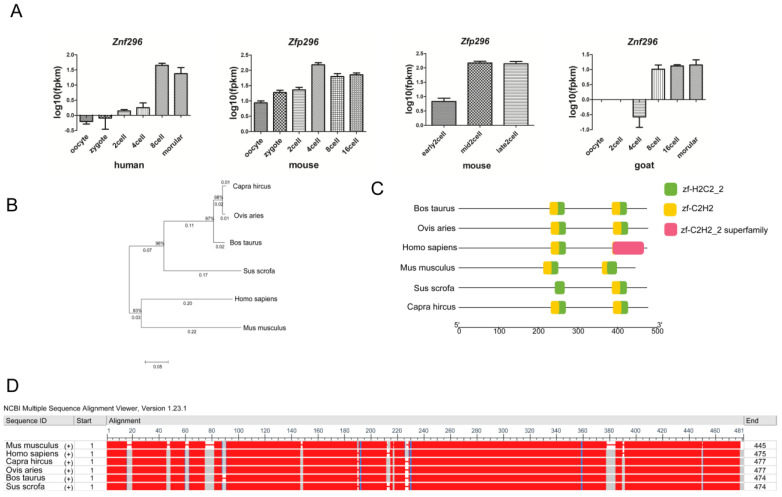
The expression of *Zfp296* in human, mouse, and goat early embryos and its species conservation. (**A**) Expression of *Zfp296* in human, mouse and goat early embryos; (**B**) DNA sequence alignment of *Zfp296* in different species; (**C**) identification of conserved domains of ZFP296 in human, mouse, goat, sheep, bovine, and pig; (**D**) protein sequence alignment of ZFP296 from human, mouse, goat, sheep, bovine, and pig (red means the same sequence).

**Figure 2 ijms-24-11377-f002:**
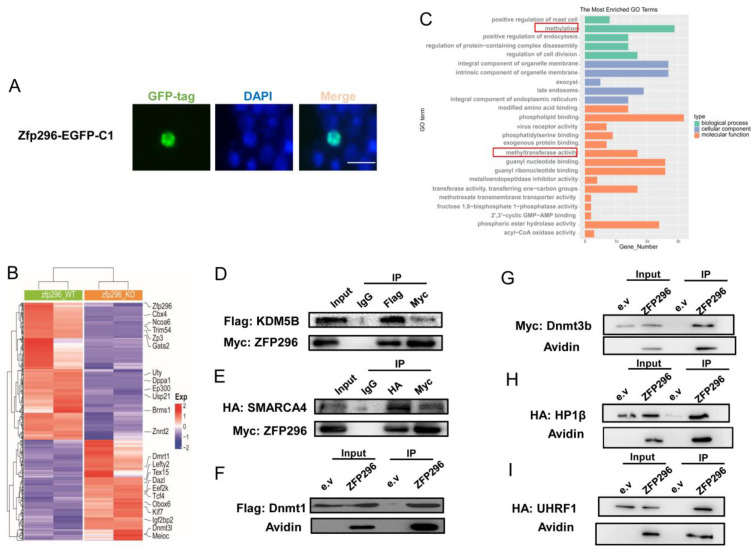
Cellular localization and interacting proteins of ZFP296. (**A**) ZFP296 localizes to the nucleus in F9 cells; bar, 10 μm; (**B**) heatmap of differential genes (*padj*  < 0.05, |log2(fold change)|≥ 1) in *Zfp296* KO mESCs compared to wild type; (**C**) GO enrichment of differential genes in *Zfp296* KO mESCs; (**D**–**I**) Myc-tagged ZFP296 (**D**,**E**) or Avidin-tagged ZFP296 (**F**–**I**) were co-expressed with Flag-tagged KDM5B, HA-tagged SMARCA4, Flag-tagged DNMT1, Myc-tagged DNMT3b, HA-tagged HP1β, or HA-tagged UHRF1 in HEK-293T cells.

**Figure 3 ijms-24-11377-f003:**
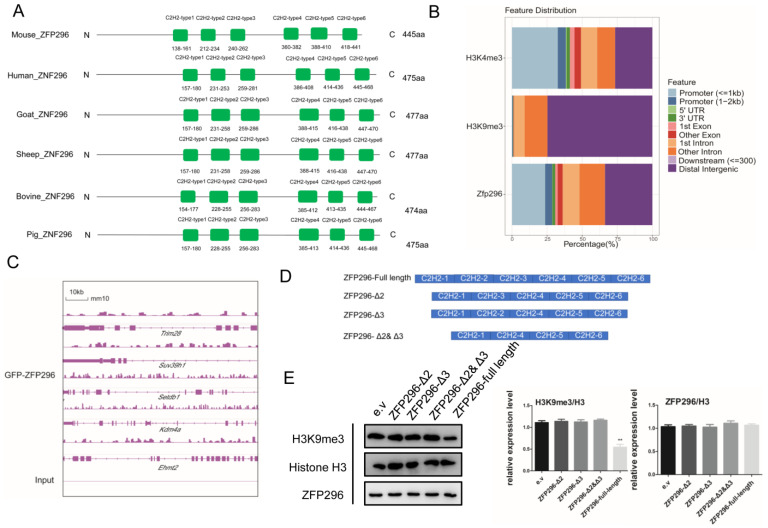
Relationship of *Zfp296* with H3K9me3. (**A**) The C2H2-TYPE domains of ZFP296 in different mammals; (**B**) genomic distribution of *Zfp296*, H3k4me3, and H3k9me3 in mESCs; (**C**) depiction of ZFP296 enriched in *Trim28*, *Suv39h1*, *Setdb1*, *Kdm4a*, and *Ehmt2* by ChIP-Seq analysis; (**D**) deletion mutants of *Zfp296* used in this study; (**E**) the effect of C2H2-TYPE2 and C2H2-TYPE3 on H3K9me3 (** *p* < 0.01). Data were mean ± s.d., *n* = 3 independent experiments, one-tailed Student’s *t*-test.

**Figure 4 ijms-24-11377-f004:**
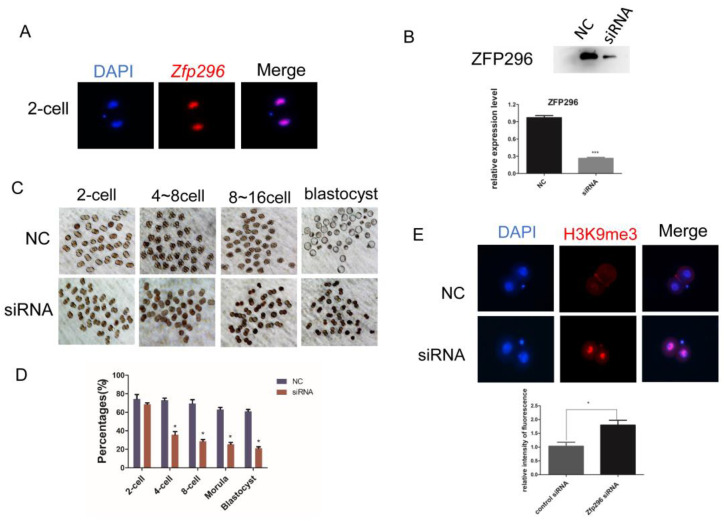
The effect of *Zfp296* on early development of mouse embryos. (**A**) *Zfp296* localized to the nucleus in 2-cell; (**B**) Western blotting of interference with *Zfp296* expression (*** *p* < 0.001); (**C**,**D**) knockdown of *Zfp296* arrested the development of mouse early embryos (* *p* < 0.05); (**E**) interference with *Zfp296* expression leads to upregulation of H3K9me3 modification in the nucleus; bar, 25 μm (* *p* < 0.05). Data in (**B**–**E**) were mean ± s.d., n = 3 independent experiments, one-tailed Student’s *t*-test.

**Table 1 ijms-24-11377-t001:** Primer sequences of different mutants of *Zfp296*.

Different Mutants of *Zfp296*	Forward	Reverse
*Zfp*296 full-length	CCCGAATTCAGATGTCCCGCCGCAAGGC	CGCGGTACCTCAGGCCATCTCTGGGTG
*Zfp*296-Δ2	GTGTTCCTATGCCTGCGCTCAGAGCAGCAAGCTCAACAG	CTGTTGAGCTTGCTGCTCTGAGCGCAGGCATAGGAACAC
*Zfp*296-Δ3	TCCCACACTGGTGAGCGACCCTACGGCAGCTGGCACCCG	CGGGTGCCAGCTGCCGTAGGGTCGCTCACCAGTGTGGGA
*Zfp296*-Δ2&Δ3	TGAGCAGTGCTGCCCGGCGGAGCCCCGGCAGCTGGCACCCGGGA	TCCCGGGTGCCAGCTGCCGGGGCTCCGCCGGGCAGCACTGCTCA

**Table 2 ijms-24-11377-t002:** Primer sequences of vectors for co-immunoprecipitation.

Different Vectors	Forward	Reverse
*Zfp296*-Bio	CGCGAATTCATGTCCCGCCGCAAGGCC	CCCGGATCCGGCCATCTCTGGGTGCTT
Flag-*Kdm5b*	GGGGTACCATGGAGCCGGCCACCACGCT	GCTCTAGATTACTTTCGGCTTGGTGCGTCCTTC
HA-*Smarca4*	AAGGAAAAAAGCGGCCGCATGTCTACTCCAGACCCACCCTTGG	AAGGAAAAAAGCGGCCGCTCAGTCTTCCTCACTGCCACTTCC
Flag-*Dnmt1*	GGGGTACCCAATGCCAGCGCGAACAGCTCCAGCCC	GCTCTAGAGTCCTTGGTAGCAGCCTCCTCTTTT
Myc-*Dnmt3b*	GGAATTCGAATGGGGAAAAAGCAAAACAAGAAGAA	GGGGTACCCTAATTCTTGTCGTCTTTTTTGT
HA-*Uhrf1*	GGAATTCGAATGTGGATCCAGGTTCGAACTATGG	GGGGTACCTCACCGGCCGCTGCCATAGCCAGG
HA-*Hp1β*	TGAGCAGTGCTGCCCGGCGGAGCCCCGGCAGCTGGCACCCGGGA	TCCCGGGTGCCAGCTGCCGGGGCTCCGCCGGGCAGCACTGCTCA

## Data Availability

Additional data are provided in the Appendix A.

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
