# Peer review of "The Novel Role of Zfp296 in Mammalian Embryonic Genome Activation as an H3K9me3 Modulator"

_ijms, 2023, doi:10.3390/ijms241411377_

Round 1

Reviewer 1 Report

The presented data are original and are bringing new knowledge. The results are clearly presented, but suffer from poor English language quality. Especially in Results chapter this results in reduced intelligibility of the text. Figure and tables are well arranged, discussion has no major shortcomings, references are appropriate. I suggest the editing of the English language. Apart from that, I have only a few minor comments.

·        Distinguish more clearly which experiments were carried out in your laboratory and when you only analyzed the results obtained by another research group.

·        Line 212 – This is rather figure 1A and comparison to early 2c stage than to zygote.

·        Explain the abbreviations used (especially in figures).

·        The localization of Zfp296 should be shown not only in 2-cell stage but throughout the whole preimplantation development from MII stage oocyte to at least 8-cell stage.

Especially in the Results chapter, the text is difficult to understand due to bad English.

Author Response

Dear reviewer

Thank you for your advice

Based on your suggestions, the English language was revised by an established expert in English, especially in the Results chapter.

We cited the  references in Results section and Materials and methods section of the results from re-analyzed RNA-seq and ChIP-seq. And we also emphasized RNA-seq and ChIP-seq were published in the Results section.

Thank you very much for your pointed the mistake of line 212 and we corrected.

We also added the table S4 to clarify the  abbreviations in the manuscript and the localization of Zfp296 from MII stage oocyte to 8-cell stage in supplementary  materials.

Thank you for all your comments, it makes us improve a lot.

Best regards

Reviewer 2 Report

22nd June, 2023

Review of Manuscript ID: ijms-2462879 , by L. Gao et al., entitled: “The novel role of Zfp296 in mammalian embryonic genome activation as a H3K9me3 modulator that is intended for publication in International Journal of Molecular Sciences

(The Microsoft Word file as Reviewer Attachment for Manuscript ID ijms-2462879 Int. J. Mol. Sci. 22nd June 2023 has also been added)

The purpose of the manuscript is to determine the role played by Zfp296 gene at the early stages of embryogenesis related to embryonic genome activation (EGA) and the changes occurring in the transcriptomic profiles and epigenomic signatures. Considering three mammalian species (humans, mice, goats), the Authors have showed that ZFP296 gene expression is significantly elevated during the EGA processes followed by Zfp296-mediated interaction with other proteins involved in histone and DNA methylation. Moreover, Zfp296 gene has been found to be responsible for regulating the H3K9me3 levels and has been found to impact on the early embryonic development.

The paper is valuable and relatively well written in English, the topics of the manuscript falls within the scope of the publisher. It is worth highlighting an abundant methodological workshop based on the use of a wide variety of methods from the fields of bioinformatics analyses, molecular genetics, genomics, transcriptomics and immunocytochemistry. The Authors have selected the relevant methods for statistical analysis of the results. This enabled to critically evaluate the results achieved by the Authors as compared to the studies of other investigators.

But, in my opinion, the following points should be considered prior to the acceptance of manuscript for publication as has been detailed below:

1) It would be appropriate to add a separate Abbreviations section to clarify the wide range of intra-textual abbreviations used by the Authors.

2) The Conclusions sections is deprived of important future goals and research directions resulting from the studies conducted so far. Therefore, the following paragraphs and in-text citations of 7 detailed References have to be added by the Authors at the end of the Conclusions section as shown below:

Collectively, identifying and exploring the molecular and epigenetic factors determining inter-transcriptomic and inter-proteomic communication during EGA processes in human, murine and caprine embryos might contribute to improvement of producing the high-quality embryos by using a wide spectrum of assisted reproductive technologies (ARTs) and experimental embryology techniques (EETs) not only in the above-indicated three species, but also in other mammalian species. The aforementioned ARTs and EETs include such modern methods as in vitro fertilization either by gamete co-incubation or by intracytoplasmic sperm injection [73–76] and somatic cell cloning [77–79].

References:

[73] Zhang, X.; Lian, F.; Liu, D. Comparison of IVF/ICSI outcomes in advanced reproductive age patients with polycystic ovary syndrome and advanced reproductive age normal controls: a retrospective cohort study. BMC Pregnancy Childbirth 2023, 23, 440. doi: 10.1186/s12884-023-05732-0.

[74] Chu, D.; Wang, H.; Fu, L.; Zhou,  W.; Li, Y. A method to improve embryo development potential when fertilization is delayed in mice. Syst. Biol. Reprod. Med. 2020, 66, 337–341. doi: 10.1080/19396368.2020.1785041.

[75] Menéndez-Blanco, I.; Soto-Heras, S.; Catalá, M.G.; Roura, M.; Izquierdo, D.; Paramio, M.T. Effect of crocetin added to IVM medium for prepubertal goat oocytes on blastocyst outcomes after IVF, intracytoplasmic sperm injection and parthenogenetic activation. Theriogenology 2020, 155, 70–76. doi: 10.1016/j.theriogenology.2020.06.008.

[76] van der Weijden, V.A.; Schmidhauser, M.; Kurome, M.; Knubben, J.; Flöter, V.L.; Wolf, E.; Ulbrich, S.E. Transcriptome dynamics in early in vivo developing and in vitro produced porcine embryos. BMC Genomics 2021, 22, 139. doi: 10.1186/s12864-021-07430-7.

[77] Skrzyszowska, M.; Samiec, M. Generating Cloned Goats by Somatic Cell Nuclear Transfer-Molecular Determinants and Application to Transgenics and Biomedicine. Int. J. Mol. Sci. 2021, 22, 7490. doi: 10.3390/ijms22147490.

[78] Wakayama, S.; Terashita, Y.; Tanabe, Y.; Hirose, N.; Wakayama, T. Mouse Cloning Using Outbred Oocyte Donors and Nontoxic Reagents. Methods Mol. Biol. 2023, 2647, 151168. doi: 10.1007/978-1-0716-3064-8_7.

[80] Samiec, M.; Skrzyszowska, M. Preimplantation developmental capability of cloned pig embryos derived from different types of nuclear donor somatic cells. Ann. Anim. Sci. 2010, 10, 385398.

3) The References section has to be prepared according to the requirements of International Journal of Molecular Sciences.

In conclusion, the paper can be accepted for publication in International Journal of Molecular Sciences, provided that the Authors will have taken into consideration all recommendations of the Reviewer and, as a consequence,  the minor revision of the manuscript will have been made.

The manuscript has to be only slightly revised for English language and style.

Author Response

Dear reviewer

Thank you for your advice.

Based on your suggestions,we added the paragraphs at the end of Conclusions section and cited the seven  detailed references as your recommendation.

And we also added the table S4 to clarify the  abbreviations in the manuscript and the English language was revised by an established expert in English.

Best regards

Reviewer 3 Report

Embryo Genome Activation (EGA) is a crucial step that takes place early in the development of an embryo. Understanding the roles of key regulators in the early stages of embryonic development is essential for uncovering the complexity and systematic nature of this process. The timing of EGA varies among different species, and there is significant interest in studying the changes in biparental transcript and protein levels, as well as alterations in epigenetic modifications. Recent studies have revealed that in mice, the major EGA occurs at the 2-cell stage, while a minor activation happens at the zygote stage. In humans, EGA begins at the one-cell stage and mainly occurs at the 8-cell stage. In bovine and goat embryos, EGA occurs between the 8-cell and 16-cell stages. Although the timing and pattern of maternal mRNA degradation and embryonic transcription activation differ across species, they are still conserved to some extent.

The manuscript is well-written, prooviding an excellent overview of EGA, the changes in epigenetic modifications during this process, and the proteins involved. The study's objectives are clearly stated, and the methods employed, mostly focusing on mouse embryos, are up-to-date and well-described.

The study's results demonstrate that RNA-Seq analysis of early embryos from humans, mice, and goats shows higher expression of Zfp296 during the EGA stage compared to the oocyte stage in all three species. Furthermore, Zfp296 is conserved across species, including humans, mice, goats, sheep, pigs, and bovines. The authors also discovered that ZFP296 interacts with several epigenetic regulators, namely KDM5B, SMARCA4, DNMT1, DNMT3B, HP1β, and UHRF1. The C2H2-TYPE2 and C2H2-TYPE3 domains of ZFP296 co-regulate the modification level of H3K9me3. Through ChIP-seq analysis, the authors demonstrated that ZFP296 is enriched in Trim28, Suv39h1, Setdb1, Kdm4a, and Ehmt2 in mESCs genome. Additionally, the knockdown of Zfp296 expression in mice resulted in the early developmental arrest of mouse embryos.

The discussion of the results is comprehensive and displays a high level of expertise. The literature cited is well selected and covers all important aspects of this study.

Formularbeginn

Formularende

Author Response

Thank you for your recognition and I appreciate it very much!

Reviewer 4 Report

Although the paper has significant findings, the abstract does not give a complete picture of the paper aims and what the author achieved from these aims. The author re-analyzed published RNA-Seq data related to the comparison of oocyte genome to that of the preimplantation embryos. However, what is written in the abstract doesn’t clearly state this. Instead, it may give a different impression that the RNA-Seq itself is part of the study. I am not against reanalysis of work published by researchers, but the authors should revise the following sentence:

Line 14-15: In this study, the RNA-Seq analysis of human, mouse and goat early embryos revealed that Zfp296- expression is higher at the EGA stage than the oocyte stage in all three species (padj < 0.05 |log2(foldchange)| ≥ 1).

The abstract lacks a concluding statement.

The writing style, especially that of the methodological part, should be re-checked so that it appears in a more coherent way.

The study rationale should be mentioned in the abstract and stressed inside the paper.

The introduction is lengthy and distracting. Authors are advised to revise it, trim paragraphs, and focus mainly on relevant information.

Except for very well-known abbreviations, all abbreviations should be written in full on the first mention. Examples are EGA, Zfp296, KDM5B, SMARCA4, DNMT1, DNMT3B, HP1β, UHRF1, SCNT

Line 16-17: Subsequently, we found that Zfp296 of human, mouse, goat, sheep, pig and bovine are conserved cross species. Change to: Subsequently, we found that Zfp296 is conserved a cross human, mouse, goat, sheep, pig, and bovine.

Line 16: padj < 0.05 replace padj with adjusted p-value.

Line 27: In early embryogenesis .. change to During early embryogenesis.

Line 35: to reveal the complexity and systematic of ..  revise the use of the word “systematic”.

Line 45: not recommended to start a paragraph with “Furthermore”.

Line 47: For instance, Different types > different types.

Line 45-71: This paragraph is too long. The author are advised to trim or split into two.

Line 99: WT part of RNA-Seq … what do you mean by WT? Also, I didn’t find the reference # 34 having any Seq analysis, just RT-PCR, please revise! You may list these citations next to each species so that confusion is minimized.

Line 110: Please clear in the manuscript what you meant by “muscled”.

Line 118-119: Figure 1. RNA-seq analysis of human, mouse, and goat early embryos and the conserved Zfp296 in different species. This title should be changed to reflect the nature of the work.

Line 201 Figure 4. The effect of Zfp296 on early embryonic development. change to “The effect of Zfp296 on early development of mouse embryos”.

Line 308: Use correct grammar.

Line 462: In conclusion, we not only shown that the expression of Zfp296 was significantly upregulated … showed!

Table S1. If possible, add the directions (up and down) and fold change of these genes.

The English of the paper requires moderate revision.

Author Response

Dear reviewer

Thank you for your advice

Based on your suggestions, we corrected the mistakes and revised the sentences that you pointed.

We added the concluding statement and the study rationalein the abstract stresses the rationalein in the introduction. Meanwhile, we revised the introduction more short and focused on relevant informatio. 

Based on your suggestions, we written in full of abbreviations on the first mention. But the abstract will be too long if we wrritten in full of KDM5B, SMARCA4, DNMT1, DNMT3B, HP1β, UHRF1. So we wrritten then in full on the main text. And we also added the table S4 to clarify the abbreviations in the manuscript.

WT means wild type of RNA-seq and we changed it back in the paper. Reference # 34 was a mistake and we deleted it from the paper. We apologized for it.

The list of citations to species are as following

Reference # 19 mouse

Reference # 35 goat

Reference # 36 human

The  “muscled” came from the way aligned DNA sequences,it is called Muscle. We written in this way is not appropriate and we changed it to "aligned". We apologized again.

We added the directionsand fold change of genes in Table S1

As your advice, the English language was revised by an established expert in English, especially in the results and methodological part.

Thank you for all your comments, it makes us improve a lot.

Best regards

Round 2

Reviewer 4 Report

The authors have clarified my concerns and the manuscript is now improved. However, several typing errors are still encountered, examples are shown below.

Line 23: leaded ….. and and!

Line 24: a H3K9me3 modulator > an H3K9me3 modulator

Line 41, 106: cross: across ?

Line 145: highly species conservation > high species conservation

Line 152: are shared > shared or are sharing

Line 155: Figures. 1A  > Figure (also 195, 200, 202, 238, 241, 247, 250)

Line 165-166: it is species conserved > its species conservation

Line 196: to compared with >  to compare with ?

Line 209: dramaticly > dramatically ?

Line 241: Mosue ? ?

Line 415-417: We downloaded and re-analyzed the wildtype part of RNA-seq from mouse (GSE129731), human (GSE36552) and goat early embryos (GSE129742): since these datasets are part of published papers, cite the paper next to each GSE#.

The English of the manuscript needs to be thoroughly revised before further consideration.

Author Response

Dear reviewer

Thank you for you pointed out the mistakes in the manuscript. We corrected them all and revised the manuscript thoroughly.

Based on your advice, we cited the paper next to each GSE#

Thank you very much!

Best regards